# The Antagonizing Role of Heme in the Antimalarial Function of Artemisinin: Elevating Intracellular Free Heme Negatively Impacts Artemisinin Activity in *Plasmodium falciparum*

**DOI:** 10.3390/molecules27061755

**Published:** 2022-03-08

**Authors:** Pan Zhu, Bing Zhou

**Affiliations:** 1State Key Laboratory of Membrane Biology, School of Life Sciences, Tsinghua University, Beijing 100084, China; zhupan6121@163.com; 2Shenzhen Institute of Synthetic Biology, Shenzhen Institute of Advanced Technology, Chinese Academy of Sciences, Shenzhen 518055, China

**Keywords:** heme, artemisinins, hemozoin, triarylimidazole 14c, heme oxygenase

## Abstract

The rich source of heme within malarial parasites has been considered to underly the action specificity of artemisinin. We reasoned that increasing intraparasitic free heme levels might further sensitize the parasites to artemisinin. Various means, such as modulating heme synthesis, degradation, polymerization, or hemoglobin digestion, were tried to boost intracellular heme levels, and under several scenarios, free heme levels were significantly augmented. Interestingly, all results arrived at the same conclusion, i.e., elevating heme acted in a strongly negative way, impacting the antimalarial action of artemisinin, but exerted no effect on several other antimalarial drugs. Suppression of the elevated free heme level by introducing heme oxygenase expression effectively restored artemisinin potency. Consistently, zinc protoporphyrin IX/zinc mesoporphyrin, as analogues of heme, drastically increased free heme levels and, concomitantly, the EC_50_ values of artemisinin. We were unable to effectively mitigate free heme levels, possibly due to an unknown compensating heme uptake pathway, as evidenced by our observation of efficient uptake of a fluorescent heme homologue by the parasite. Our results thus indicate the existence of an effective and mutually compensating heme homeostasis network in the parasites, including an uncharacterized heme uptake pathway, to maintain a certain level of free heme and that augmentation of the free heme level negatively impacts the antimalarial action of artemisinin. Importance: It is commonly believed that heme is critical in activating the antimalarial action of artemisinins. In this work, we show that elevating free heme levels in the malarial parasites surprisingly negatively impacts the action of artemisinin. We tried to boost free heme levels with various means, such as by modulating heme synthesis, heme polymerization, hemoglobin degradation and using heme analogues. Whenever we saw elevation of free heme levels, reduction in artemisinin potency was also observed. The homeostasis of heme appears to be complex, as there exists an unidentified heme uptake pathway in the parasites, nullifying our attempts to effectively reduce intraparasitic free heme levels. Our results thus indicate that too much heme is not good for the antimalarial action of artemisinins. This research can help us better understand the biological properties of this mysterious drug.

## 1. Introduction

Malaria remains one of the major threats to global public health; it is estimated to have caused 241 million cases and 627,000 deaths worldwide in 2020 [1]. Malaria is caused by parasites of the genus *Plasmodium*, especially by *Plasmodium falciparum*, which is transmitted by the *Anopheles* mosquito. *P. falciparum* strains have developed resistance to most of the commonly used antimalarials, including quinine, chloroquine and sulfadoxine-pyrimethamine [2,3,4]. Artemisinin is a sesquiterpene lactone derived from the Chinese herb *Artemisia annua* [5,6], and a number of derivatives, such as dihydroartemisinin, artemether and artesunate, have been synthesized. These semisynthetic compounds (*Artemisinins*) exhibit improvement over artemisinin not only in solubility but also efficacy [7]. At present, artemisinin-based combination therapy(ACT) is the first-line treatment for uncomplicated malaria, which combination therapy has helped prevent recrudescence after use of artemisinins and to delay the acquisition of drug resistance. Artemisinin-resistant malaria, which is clinically defined by manifestation of slow parasite clearance, emerged in southeast Asia several years ago [8,9,10,11]. Mutations in one gene, K13, were discovered to be connected to the reduced artemisinin sensitivity in some patients [11]. These mutations led to corresponding in vitro change in the ring-stage survival assay (RSA_0–3 h_) of artemisinins [10,12].

Generating cytotoxic radical species via cleavage of the endoperoxide bridge is essential for the activity of artemisinin [13]. The activator of artemisinin is controversial. Some studies suggested that iron (Fe^2+^) might play an important role in the activation of artemisinin [14]. Some biological evidence supported the hypothesis that ferrous iron (Fe^2+^) is involved in the action of artemisinin [14,15,16]; iron chelator-like deferoxamine (DFO) antagonizes with artemisinin [17]. Another possible activator is heme, an abundant form of iron within the cell. Heme can react readily with artemisinin in vitro [18,19,20], as well as in vivo [21], and heme–artemisinin adducts have been isolated and identified by high-performance liquid chromatography (HPLC) [18]. The caveat, of course, is whether this reaction truly underlies the physiological antimalarial action of artemisinins.

Several previous pieces of work tried to correlate heme metabolism to the antimalarial activity of artemisinins [22,23]. Few of them, however, took into account and determined the free heme levels. Malarial parasites gobble up large amounts of hemoglobin during the erythrocyte stage and digest the protein to nurture themselves. The concomitant release of heme is highly toxic, and the parasites detoxify it through hemozoin formation. Hemin (oxidized heme) in this form is inert and largely inaccessible physiologically. Our previous work with the Baker’s yeast *Saccharomyces cerevisiae* exploring the action of artemisinin indicated, surprisingly, that heme is primarily a negative regulator of artemisinin’s antimitochondrial function; it reacts with heme and consume its resources [24]. However, it remains to be seen whether this is the case in malarial parasites. In this work, we tried to manipulate intracellular free heme levels in *P. falciparum* by using a range of biochemical and genetic strategies to test how labile heme alterations might impact the action of artemisinin.

## 2. Results

### 2.1. Extracellular Hemin Reduced the Antimalarial Activity of Artemisinin

Hemin has been shown to inhibit the action of artemisinin in Baker’s yeast (Appendix A). To study the interaction of heme and artemisinin in *Plasmodium falciparum*, we first tried to investigate whether exogenous heme (hemin) could affect the antimalarial effect of artemisinin. We reasoned that the external hemin might find ways inside the cells and increase intracellular heme levels. This was confirmed with intraparasitic heme measurements (Appendix A). The addition of hemin to the medium decreased the sensitivity of *P. falciparum* to artemisinin (Figure 1), whereas the addition of the corresponding solvent (0.1 M NaOH) had little effect (Appendix A). The more hemin added, the more reduction in the antimalarial activity of artemisinin. In our assays, the EC_50_ value of artemisinin in *P. falciparum* was 22.23 ± 1.13 nM and was increased to 90.59 ± 6.70 nM/156.95 ± 10.35 nM with 5 μM/10 μM hemin, respectively (Figure 1A). The parasitemia of *P. falciparum* was similarly affected. Within 72 h after the drug treatment, hemin greatly reduced the activity of artemisinin’s action on parasites (Figure 1B). Developmental inhibition of *P. falciparum* by artemisinin was also negatively impacted by hemin (Figure 1C). We then repeated the experiments with dihydroartemisinin (DHA) and obtained similar results (Figure 1D). EC_50_ values are shown in Appendix A. One caveat of this strategy is that hemin might be reduced by the media and react with artemisinin before it can reach the parasites or get inactivated prior to its action. To partially address this concern, prior treatment of parasites with hemin followed by washing also decreased ART potency (Appendix A). Nevertheless, these strategies did not address the concern that red blood cells (RBC) may also be affected by external hemin supplementation. In other words, it could be argued that external hemin might change the RBC intracellular heme level, and the heme could react with artemisinin there before it reaches the parasites. Indeed, it has been reported that heme in RBC could negatively affect artemisinin activity [25].

### 2.2. Inhibiting Hemozoin Formation Antagonized Artemisinin’s Action

To avoid the concomitant effect on RBC while affecting the parasites, we had to resort to more specific means of modulating the intracellular heme homeostasis of the parasites. In an effort to increase intraparasitic heme levels, we first added 5-aminolevulinic acid (ALA), the precursor of heme biosynthesis, to the culture and measured the sensitivity of the parasites to artemisinin. No change in EC_50_ value of artemisinin was observed (Appendix A). Extracellular ALA did increase levels of heme precursors, such as coproporphyrinogen III (CPP) and protoporphyrin IX (PPIX), but had little effect on the heme [26]. Therefore, it seems that the addition of ALA was not able to increase the free heme concentration. We subsequently tested some hemozoin-inhibiting drugs active against parasites, such as triarylimidazole14c (hereafter 14c). According to the literature, it could enhance free heme levels in parasites [27]. This was indeed the case in our hands, as reported in Figure 2A. We then measured the artemisinin sensitivity of *P. falciparum* in the presence of moderate levels of 14c (3 to 5 M). EC_50_ of artemisinin was obviously increased with 14c (Figure 2B). Likewise, this also happened to DHA (Figure 2C). We used 14d as the control, which is another triarylimidazole and analogue of 14c (Figure 2D), but could not enhance the free heme level [27]. Consistently, 14d had little effect on EC_50_ of ART (Figure 2E). EC_50_ values are shown in Appendix A. To ascertain whether increasing the 14c concentration could obtain more robust antagonizing effects, we used higher levels of 14c. Unfortunately, under long incubation, 14c is toxic, presumably due to its hemozoin inhibitory and heme-enhancing effect. We therefore decided to adopt a short-pulse assay to test the sensitivity of artemisinin with higher concentrations of 14c (Appendix A). For early ring-stage parasites, EC_50_ of artemisinin significantly increased from 31.25 ± 4.67 nM to about 180 nM in the presence of 15 M of 14c (Figure 2F). Consistently, the free heme level was enhanced in the presence of 15 M of 14c (Appendix A). These data indicated that enhancing the heme level in *P. falciparum* could negatively alter the antimalarial activity of artemisinin.

### 2.3. Heme Oxygenase Suppressed the Antagonizing Effect of 14c on Artemisinin’s Action

The aforementioned experiments indicate that a dramatic increase of heme levels could happen with the addition of the hemozoin formation-inhibitor drug 14c, with a concomitant decrease in artemisinin activity. It still could be argued that there might be another unknown effect associated with 14c that truly underlies the change in artemisinin effectiveness, i.e., 14c might be associated with another biological effect, and its antiartemisinin effect may not be due to its effect on heme. To exclude this possibility, we introduced heme oxygenase (HO) into the parasites. Within the cell, HO normally breaks down heme to biliverdin and detoxifies extra heme (Figure 3A). In the malarial parasites, no HO is known. An HO homologue in *P. falciparum* could bind heme and protoporphyrin IX but could not degrade heme due to lack of a key His, a heme-coordinate residue [28]. Malarial parasites normally cope with excess heme through hemozoin formation. If the effect of 14c on artemisinin occurred through an increase in heme, we expected that exogeneous HO would be able to neutralize this action. Heme oxygenase from *Arabidopsis thaliana*, which is expected to be evolutionarily closer to malaria parasites than animals [28], was introduced to *P. falciparum* to make an HO transgenic parasite strain. HO expression by itself had no significant effect on artemisinin under normal conditions (Figure 3B). The normal function of HO is to detoxify excess heme, and unsurprisingly, free heme levels in the HO transgenic parasites had no significant change compared with the wild-type parasites, 3D7 (Figure 3C). As stated, when hemozoin inhibitor 14c was added, parasitic growth was suppressed. This suppression was greatly reversed by HO (Figure 3D), reaffirming that the toxicity of 14c mainly arises from an increase in heme levels. Likewise, whereas 5 μM 14c could enhance heme level in the parasites by severalfold and significantly increase the value of artemisinin EC_50_ in wild-type *P. falciparum*, HO expression strongly neutralized the increase in heme (Figure 3E), as well as the effect of 14c on artemisinin’s action (Figure 3F). EC_50_ values are shown in Appendix A. As a further control, we used an irrelevant protein EGFP to show that the observed effect is specific to HO and not to a random protein; the EGFP expression strain exhibited similar phenotypes to the wild-type strain (Appendix A). Expression and activity of HO were confirmed by real-time quantitative RT-PCR (Appendix A) and by measurement of biliverdin formation via LC-MS/MS (Appendix A). The biliverdin level in the HO-expression strain was not much different than that of 3D7 (Figure 3G). However, the biliverdin level was greatly enhanced with 14c in the HO-expression strain (Figure 3H), whereas at the same time, 14c had no significant effect on biliverdin in 3D7 (Appendix A). This unequivocally indicates that 14c’s effect on artemisinin acts through free heme regulation and not other possible unrelated effects of 14c.

### 2.4. Enhancing Heme Levels by Extracellular Heme Analogue also Reduced the Antimalarial Activity of Artemisinin

ZnPPIX, an analogue of heme (Figure 4A), could bind heme crystals to inhibit the process of crystallization in *P. falciparum* [29,30]. This prompted us to investigate artemisinin sensitivity of the parasite in the presence of this chemical. The EC_50_ value of artemisinin in 3D7 increased by more than twofold in the presence of 10 μM ZnPPIX compared with that of DMSO (Figure 4B). We then measured to what extent the free heme content might be affected. In 3D7, ZnPPIX increased free heme levels by more than sixfold as compared with the DMSO control (Figure 4C). Meanwhile, it inhibited hemozoin formation (Appendix A), as reported in [30], and the growth of parasites (Appendix A). ZnMP, another fluorescent analogue of heme, also had a similar effect (Figure 4D). EC_50_ values are shown in Appendix A. This also happened to DHA (Figure 4E). ZnMP offered the advantage of being able to follow its route when entering the cell. When ZnMP was added to the parasitic culture, an obvious signal was observed in the parasite within an hour (Figure 4F). Of note is that the was fluorescence predominantly located within the parasites, suggesting that a robust heme uptake pathway likely exists in the parasite. These data are consistent with the notion that enhanced heme levels decrease the activity of artemisinin in *P. falciparum.*

### 2.5. Inhibiting de Novo Heme Biosynthesis and Hemoglobin Degradation Had Little Effect on Free Heme and the Action of Artemisinin

We next sought other ways that might modulate or especially mitigate free heme levels. One obvious strategy is to regulate heme levels by specific chemicals that could repress heme biosynthesis or, alternatively, to inhibit hemoglobin degradation to prevent heme release. Heme de novo synthesis could be inhibited by succinylacetone (SA) [26], an analogue of 5-aminolaevulinic acid (ALA) (Figure 5A,B). SA had little effect on the growth of parasites (Appendix A) and the EC_50_ value of artemisinin against *P. falciparum* (Figure 5C). Since de novo heme synthesis in the erythrocyte stage is dispensable for the parasite [31], we figured that it might be because an alternative pathway compensated the heme production. Free heme is also generated via hemoglobin degradation in the food vacuoles of parasites. Two families of proteases, namely cysteine protease falcipains (falcipain 2/2′) and aspartic protease plasmepsins (plasmepsin II and IV), play important roles in executing this function [32]. *N*-[*N*-(L-3-Trans-carboxirane-2-carbonyl)-L-leucyl]-agmatine (E64) is an inhibitor of cysteine proteases that inhibits falcipain hemoglobinases, whereas pepstatin A is an inhibitor of aspartic proteases that inhibits plasmepsin hemoglobinases (Figure 5D). E64 and pepstatin A could inhibit the growth of parasites after 72h incubation (Figure 5E) and are dose-sensitive, with higher concentrations causing severer toxicity. E64 but not pepstatin A inhibited hemoglobin degradation (Appendix A). However, no significant changes in free heme and hemozoin levels were observed with these inhibitors (Appendix A). We measured effects of these inhibitors on artemisinin. Neither appreciably changed the EC_50_ values of artemisinin (Figure 5F). As stated, the de novo and hemoglobin pathways might be redundant in terms of generating free heme, and it is possible that only after the two pathways are inhibited could free heme be increased. We subsequently combined the use of inhibitors in order to simultaneously repress the heme synthesis pathway and the hemoglobin degradation pathway. SA had no interaction with E64 or pepstatin A (Appendix A), whereas E64 was synergistic with pepstatin A, with a coefficient of drug interaction (CDI) value of 0.56 (Figure 5G). Then, we measured the sensitivity of artemisinin with these inhibitors. To our surprise and disappointment, no significant change was observed, even with the application of two inhibitors (Figure 5H). EC_50_ values are shown in Appendix A. Even when three inhibitors were applied, not much effect appeared. Is this lack of effect because heme levels have a neutral effect on artemisinin’s action, or were these measures not effective in altering the free heme level? When heme levels in *P. falciparum* were measured, to our amazement, free heme levels in *P. falciparum* were not significantly altered, even after the addition of multiple inhibitors (Figure 5I). These results indicated that the free heme level inside parasites is robustly controlled and that suppression of the de novo synthesis or hemoglobin degradation pathways or both is not effective in reducing intracellular free heme, likely due to another compensating pathway. We suspect this compensating pathway might be an uncharacterized heme uptake pathway, supported by the aforementioned efficient fluorescence accumulation in the parasites after the addition of ZnMP (Figure 4F).

### 2.6. Elevating Heme Level Had Little Effect on Other Antimalarial Drugs

By this stage, we had tested various measures of modulating heme levels in *P. falciparum*, and in our hands, all measures that could significantly elevate free heme levels were found to negatively impact artemisinin’s potency. Exogeneous hemin, heme analogues and 14c all boosted free heme levels and antagonized the action of artemisinin. We then asked whether the heme increase is specific to the action of artemisinin—in other words, whether other antimalarials could be similarly affected. We first tested the effects of hemin, ZnPPIX, ZnMP and 14c on atovaquone activity. Using identical concentrations to those used in artemisinin, these chemicals had no effect on atovaquone’s action (Figure 6A–D); EC_50_ values of atovaquone with these chemicals were not noticeably changed compared with the control (Appendix A). We subsequently tested hemin, as the representative chemical, for its effects on several other antimalarial drugs. An amount of 10 μM hemin had little effect on quinine, mefloquine, proguanil and pyrimethamine (Figure 6E–I); EC_50_ values of these antimalarial drugs were minimally impacted (Appendix A). These results suggest that the interaction of heme and artemisinin is specific and that elevating free heme levels exerts a negative effect on artemisinin in inhibiting malarial parasites.

## 3. Discussion

In this work, we utilized various strategies to try to modify intracellular free heme levels within parasites. We succeeded in elevating heme levels under several circumstances, where reduced artemisinin activities were all observed. When the heightened heme level induced by hemozoin inhibition was suppressed by HO introduction, artemisinin sensitivity was restored. These results led us to conclude that heme, or at least supra-physiological levels of heme, may exert a negative role in the antimalarial action of artemisinin.

ZnPPIX and ZnMP, two heme homologues, were able to increase free heme levels and ameliorate the activity of artemisinin. Previous reports showed that ZnPPIX could inhibit crystallization of heme [29]. Therefore, the ways in which these two heme homologues act to release heme are possibly through interference of hemozoin formation and substitution of heme in heme-containing proteins. Our experiments indicated that heme homologues and hemozoin inhibitor triarylimidazole 14c are all effective in raising free heme levels accompanied with suppression of artemisinin’s potency, and HO is able to reverse these effects. We did not try HO to restore the heme levels elevated by ZnPPIX or ZnMP and artemisinin’s action out of concern that these two may inhibit HO enzymatic activity [33], making interpretation of results ambiguous.

One interesting issue is that we were unable to effectively reduce intracellular free heme levels. It seems that the parasites have multiple means of maintaining a stable heme resource. Repression of de novo synthesis plus hemoglobin degradation did not decrease free heme by much nor lead to a change in artemisinin sensitivity. In a previous study, it was likewise shown through radioisotope assays that SA suppressed heme synthesis; however, this suppression did not alter the susceptibility of the malarial parasites to artemisinin [31]. Our observation of efficient ZnMP uptake into the parasites suggests to us that the parasites have a highly effective and unknown avenue to absorb heme from the environment. This may explain why repression of heme de novo synthesis or hemoglobin degradation did not significantly alter free heme levels.

In previous work, when pepstatin A was used to treat the parasites, no drop in potency for artemisinin was observed in a short-pulse assay [22]. In another study with long-term artemisinin treatment, the inhibitor of plasmepsin hemoglobinase, Ro 40-4388, had no antagonistic interaction with artemisinin over the full erythrocytic stage [34], which is consistent with our observation. However, it was reported that cystatin protease inhibitors E64 and ALLN antagonized artemisinin’s action at the trophozoite stage in a short-pulse assay [22,23]. It is not known, however, whether this treatment altered the free heme levels for parasites at this stage. We consider that short treatment might also bring in another uncertainty, i.e., it could be argued that protease inhibitor treatment might disrupt the metabolism and make the parasites temporarily dormant, augmenting EC_50_ values, a situation similar to that of K13 mutations [35], wherein EC_50_ was increased in short-pulse assays but not under constant artemisinin presence [36]. On the other hand, although the antagonizing effect of cystatin protease inhibitors against artemisinin was interpreted to inhibit hemoglobin degradation, leading to a decrease in heme and a concomitant drop in artemisinin sensitivity, there are alternative and even opposite explanations. Hemoglobin was also reported to be reactive with artemisinin, and one report even stated that hemoglobin is more reactive with artemisinin [37], although another report says the opposite [19]. Furthermore, by degrading hemoglobin, most released heme is insulated in the form of hemozoin. This leaves open the question as to whether degrading hemoglobin causes more artemisinin activation or less, considering that hemoglobin is also reactive with artemisinin, even under a scenario wherein hemoglobin is less reactive with artemisinin.

The hypothesis of heme as the activator of artemisinin posits that malaria parasites have generally higher levels of heme than many mammalian cells, and this offers action specificity. A labile heme pool has been reported to be maintained at about 1.6 μM through the erythrocytic stage using a genetically encoded biosensor, higher than most other cell types [38], although higher free heme levels in trophozoites were also reported compared with those at other stages [39]. This action model, although supported by some correlative pieces of evidence [40,41], also raises some questions. Artemisinin kills the parasites at nM levels, but why does it need micromolar levels of heme to catalyze the activation? Furthermore, *P. falciparum* exhibits great stage-dependent differences in sensitivity to artemisinin during short drug pulse [42], and the stage sensitivity does not correlate with labile heme levels. Looking across different species, labile heme levels range from 614 to 1042 nM in IMR90 lung fibroblast cells [43] and 20 to 40 nM in *Saccharomyces cerevisiae* [44]; however, the former is highly insensitive to artemisinin, whereas the latter is sensitive to artemisinin (on non-fermentable media). Artemisinin is highly selective in killing malarial parasites, but labile heme levels in mammalian cells vary from cell to cell and, in some cases, could even approach that in malarial parasites [43]. The discrepancies between heme levels and artemisinin sensitivities across different species does not confute heme as the key underlying factor for the antimalarial action of artemisinin but does raise some concerns about attributing the antimalarial specificity to higher heme levels in the parasites. Results presented in this work do not confute heme as an activator for artemisinin’s action, since we were unbale to mitigate free heme levels. In fact, considering its indispensability, it may not be possible to entirely eliminate this factor. It could be argued that an optimal heme level optimizes artemisinin potency, and anything below or above it is not good.

Heme does play a positive role in the action of many other functions of artemisinins. In cancer cells, for example, heme elevation increases, whereas heme suppression decreases, artemisinins’ potency [45]. In the action of artemisinins against several parasites other than the malarial parasites, there are also strong indications that heme plays a facilitatory role [46]. In these reactions, EC_50_ of artemisinins are usually orders of magnitude higher than that against malarial parasites. Efficient execution of these high levels of drugs could be facilitated by elevating the levels of the activator. We explained that ROS generated by heme underly these types of actions of artemisinins and are generally non-specific, whereas in the case of malarial parasites, a more specific action exists for artemisinins. A glimpse into this extraordinary action against malarial parasites was originally obtained with Barker’s yeast studies and later confirmed with malarial parasites [24,47]. This special action appears to be specific depolarization of the mitochondrial membrane potential in the malarial parasites [48], although plasma membrane depolarization was also reported [49]. Other action models also exist. For example, it has been proposed that artemisinin reacting with FADH or flavoenzymes may disrupt the redox balance and inhibit parasites [50].

Taken together, it is becoming apparent that more than one type of action exists for artemisinins. In terms of heme and artemisinin, boosting intracellular free heme levels may augment the potency of artemisinins against many types of cells but reduce the antimalarial action of artemisinins. Heme reaction with artemisinin generates free radicals, and conceivably, this causes damage when free radicals reach a certain level. However, when artemisinin levels are low and the associated free radical level is not high, the produced free radicals pose little harm to the cell; in the case of antimalarial actions, reaction of artemisinin with the extra heme only wastes the drug—in other words, negatively impacts the action of artemisinin.

## 4. Materials and Methods

### 4.1. Materials

RPMI medium 1640 and AlbuMAX II lipid-rich BSA were obtained from Gibco Life Technologies (Grand Island, NY, USA). Artemisinin, HEPES, NaHCO_3_, hypoxanthine, sorbitol, Hemin, pepstatin A and SA were purchased from Sigma-Aldrich (St. Louis, MO, USA). E64 was purchased from Selleck (Houston, TX, USA). Dihydroartemisinin was purchased from Chengdu Okay Medicine Co., Ltd. (Chengdu, China). ZnPPIX was obtained from Alfa Aesar (Shanghai, China). ZnMP was obtained from Frontier Scientific (Logan, UT, USA). Triarylimidazole14c and 14d were gifts from Timothy J. Egan (University of Cape Town, South Africa). Hemin was dissolved in 0.1 M NaOH. ZnPPIX, ZnMP, E64, pepstatin A, 14c, 14d, atovaquone, artemisinin, dihydroartemisinin, quinine, mefloquine, proguanil and pyrimethamine were dissolved in DMSO. SA was dissolved in H_2_O.

### 4.2. Parasites Culture

*P. falciparum* (3D7) was cultured in human O^+^ erythrocytes at 37 °C under 5% CO_2_, 5% O_2_ and 90% N_2_. One liter of complete medium contained 10.4 g RPMI medium 1640 (Invitrogen), 5 g AlbuMAX II, 5.94 g HEPES, 2.2 g NaHCO_3_, 50 mg hypoxanthine and 25 mg gentamicin. Parasitemia was determined by counting the number of infected RBCs in Giemsa-stained thin blood smears. Thin blood smears were made from parasite culture, followed by methanol fixation and staining with 1:10 diluted Giemsa solution for 5 min. Ring-stage parasites were obtained through synchronization by 5% sorbitol.

### 4.3. Inhibition Assay

Synchronized ring-stage parasites were incubated with antimalarial drugs and chemicals (hemin, 14c, 14d, ZnPPIX, ZnMP, E64, Pepstatin A and SA), at 2% hematocrit and 1% starting parasitemia for 72 h. Parasites were lysed by the lysis buffer (10 mM Tris-HCl,1 mM EDTA, pH 7.5, 2% Triton X-100, 0.01% saponin) containing SYBR Green. The saponin-lysed samples were incubated in the dark at 37 °C for 30 min and then measured by a microplate fluorometer (Fluoroskan Ascent, Thermo Fisher, Waltham, MA, USA) with excitation and emission wavelengths centered at 485 nm and 538 nm, respectively. Results were normalized using controls, which only contained toxic chemicals. EC_50_ values were determined using GraphPad Prism.

### 4.4. Heme Fractionation and Measurement

Heme fractionation and measurement of heme levels were conducted as described previously [50,51]. Synchronized ring-stage parasites at 5% parasitemia and 3% hematocrit were cultured with DMSO or hemin analogue ZnPPIX or inhibitors (14c, 14d, E64, pepstatin A and SA) for 20 h. Parasites were harvested at the trophozoite stage. A proportion of 0.15% saponin-lysed samples were washed three times by PBS to remove erythrocyte hemoglobin, and pellets were resuspended by PBS. Pellets were counted by hemocytometer, and then samples were frozen at −80 °C. After thawing, 50 μL of milliQ H_2_O was added, and the plate was sonicated for 5 min in an ultrasound bath. An amount of 50 μL 0.2 M HEPES buffer (pH 7.5) was added and then centrifuged at 3600 rpm for 20 min. The supernatant was transferred to new tubes, and 50 μL 4% SDS was added, sonicated for 5 min and then incubated at 37 °C for 30 min. Then, 50 μL 0.3 M NaCl and 50 μL of 25% pyridine in 0.2 M HEPES were added (corresponding to the hemoglobin fraction). For the pellet, 50 μL H_2_O and 50 μL 4% SDS were added, and the resuspended pellet was sonicated for 5 min and then incubated at 37 °C for 30 min to solubilize the heme. Amounts of 50 μL 0.2 M HEPES, 50 μL 0.3 M NaCl and 50 μL 25% pyridine in 0.2 M HEPES were added, and then the sample was centrifuged at 3600 rpm for 20 min. The supernatant was transferred to new tubes (corresponding to the free heme fraction). The pellet was solubilized in 50 μL H_2_O and 50 μL 0.3 M NaOH, sonicated for 15 min and incubated at 37 °C for 30 min. Amounts of 50 μL 0.2 M HEPES, 0.3 M HCl and 25% pyridine in 0.2 M HEPES were added (corresponding to the hemozoin fraction). The three heme species were measured as a low-spin heme–pyridine complex, and heme levels were calculated based on the absorbance at 410 nm. The total amount of heme in each sample was quantified using a standard curve.

### 4.5. Confocal Microscopy

*P. falciparum* was cultured at 37 °C under 5% CO_2_, 5% O_2_ and 90% N_2_ in the presence of 20 μM ZnMP for 1 h, and then red blood cells were washed by PBS to remove ZnMP in the medium. ZnMP uptake was examined under a Nikon Ti-E confocal microscope equipped with a ×100 MPLAPON oil objective with the excitation wavelength at 488 nm and the emission spectrum at 550–650 nm.

### 4.6. P. falciparum Transfection

*Arabidopsis* heme oxygenase gene (AtHO) was inserted into pCC4 plasmid, a vector with a blasticidin (BSD) selectable marker. The final 849 base pairs of the AtHO sequence were PCR-amplified using the primers HO1-F (CCACATTTCGAATAAACTCGAGGCCACCATGGCGTATTTAGCTCCGAT) and HO1-R (CATATCCTTAATTAAGTCGACTCAGGACAATATGAGACGAAGT). The resulting PCR fragment was cloned into the pCC4 vector using a Gibson cloning kit.

Ring-stage *P. falciparum* 3D7 parasites at 5% parasitemia were transfected with 100 μg plasmid DNA. Erythrocytes was resuspended in Cytomix buffer (120 mM KCl, 10 mM K_2_HPO_4_/KH_2_PO_4_, 25 mM HEPES, 2 mM EGTA, 0.15 mM CaCl_2_, 5 mM MgCl_2_, adjusted pH = 7.6 by KOH) with 100 μg plasmid DNA. Then erythrocytes/DNA/Cytomix suspension were transferred into a 2 mm electroporation cuvette (Bio-Rad) and electroporated (Bio-Rad) at 0.31 kV with a capacitance of 960 μF. After electroporation, all/most cells were removed and transferred to a flask. Expression of stably-transfected HO clones were confirmed by real-time quantitative PCR, with primers qPCR-F (CAAGGTTACGAGATTCCA) and qPCR-R (AAGTAGATGTTGTAGAAGTGA).

### 4.7. LC-MS/MS Quantification of Biliverdin

The analysis was performed on an AB SCIEX QTRAP 6500 (USA) triple quadrupole mass spectrometer in SRM and positive ionization mode. Samples were prepared by 40%/60% discontinuous Percoll gradient to remove uninfected RBCs and harvest trophozoites and schizonts. Each sample was then resuspended in 100 μL methanol. LC separation was run on an XSelect HSS T3 column (2.1 × 50 mm, 2.5 μm, Waters USA) equipped with a Shimadzu Nexera X2 LC-30AD infinite binary pump using acetonitrile containing 0.05% formic acid as solvent A and water containing 0.05% formic acid as solvent B. LC parameters were as follows: flow rate, 0.3 mL/min; for gradient elution, initial conditions were 70% solvent B starting at 0 min, decreasing to 0% B at 7 min, returning to the initial state of 70% B after 2 min and ending once at the 12 min cycle. The column temperature was 30 °C. The sample was kept at 10 °C, and the injection volume was 5 μL. MS parameters were as follows: ESI ion source temperature: 500 °C; air curtain: 30 psi; collision-activated dissociation (CAD) gas settings: medium; ion spray voltage: 5500 V; ion gas 1 and 2: 50 psi. All data were analyzed with Analyst 1.6. 3 Software.

### 4.8. Statistical Analysis

The data were analyzed by Origin 8.5 and GraphPad Prism 6. Comparison was performed by unpaired *t*-test or analysis of variance (ANOVA) with the accepted *p*-value < 0.05. Statistical details are indicated in the figure legends.

## Figures and Tables

**Figure 1 molecules-27-01755-f001:**
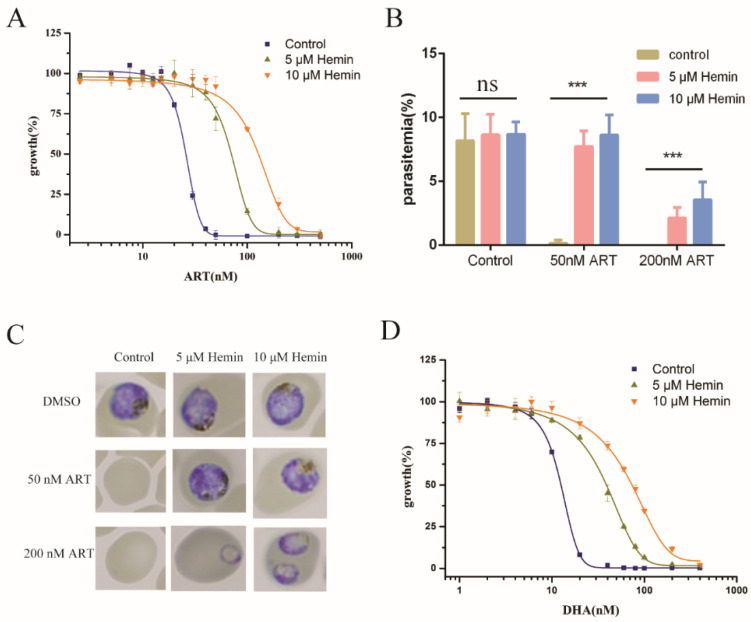
Antimalarial effectiveness of artemisinin was reduced by exogenous hemin. (**A**) Inhibitory effect of artemisinin was decreased with the addition of exogenous hemin. (**B**) Hemin antagonized with artemisinin in inhibiting parasite multiplication, suppressing the parasitemia of *P. falciparum*. (**C**) Hemin released the inhibition of artemisinin on the development of the parasites after 72 h incubation. (**D**) Similarly to artemisinin, augmenting heme levels through hemin decreased sensitivity of the parasite to DHA. ART: artemisinin, DHA: dihydroartemisinin. Data are mean ± SD of three independent experiments. The data were analyzed by GraphPad Prism. *** *p*-value < 0.0005, and ns means no significant change (*p*-value > 0.05).

**Figure 2 molecules-27-01755-f002:**
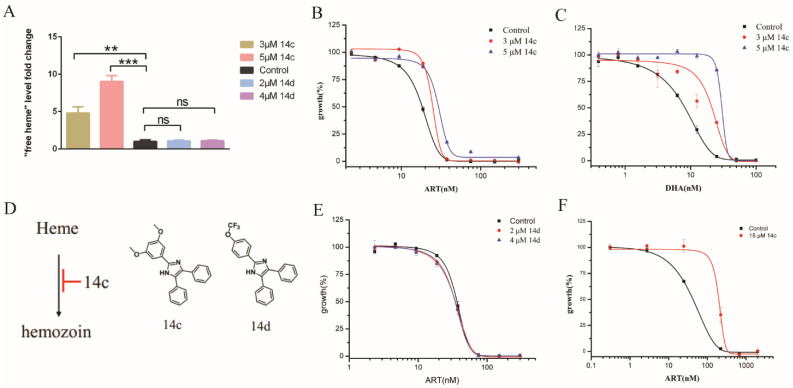
Hemozoin inhibition enhanced free heme levels and decreased sensitivity of artemisinin. (**A**) 14c but not 14d increased “free heme” levels in *P. falciparum*. Data are mean ± SD of three experiments. (**B**) EC_50_ of artemisinin was increased with 14c. (**C**) 14c obviously decreased sensitivity of parasites to dihydroartemisinin. (**D**) Chemical structure of 14c and 14d; (**E**) 14c analogue 14d had little effect on EC_50_ of artemisinin. (**F**) EC_50_ of artemisinin could also be significantly changed with higher dosages of 14c in the short-pulse assay. Early ring-stage parasites were used. Data are mean ± SD of three experiments. The data were analyzed by GraphPad Prism. ** *p*-value < 0.005, *** *p*-value < 0.0005, and ns = no significant change (*p*-value > 0.05).

**Figure 3 molecules-27-01755-f003:**
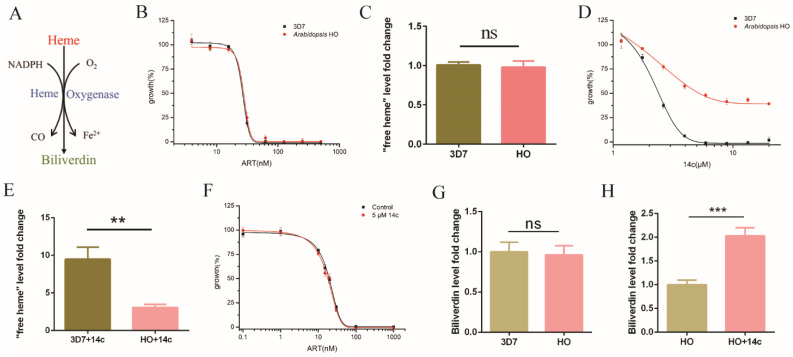
HO neutralized 14c’s effect on heme and artemisinin. (**A**) HO executes the heme breakdown function. (**B**) The introduction of HO did not affect the EC_50_ value of artemisinin in transgenic parasites. (**C**) “Free heme” level in HO transgenic parasites had no significant change with 3D7 parasites. (**D**) HO transgenic parasites were more resistant to 14c than the wild-type parasite. (**E**) Arabidopsis HO_1_ mitigated the increase in free heme by 5 μM 14c. (**F**) HO expression in *P. falciparum* neutralized the antagonizing action of 14c on artemisinin. Data are mean ± SD of three experiments. (**G**) HO expression-strain-infected RBC had a similar level of biliverdin compared with 3D7-infected RBC. Data are mean ± SD of four experiments. (**H**) Treatment with 5 μM 14c enhanced biliverdin levels in HO expression-strain-infected RBC. The data were analyzed by GraphPad Prism. ** *p*-value < 0.005, *** *p*-value < 0.0005, and ns = no significant change (*p*-value > 0.05).

**Figure 4 molecules-27-01755-f004:**
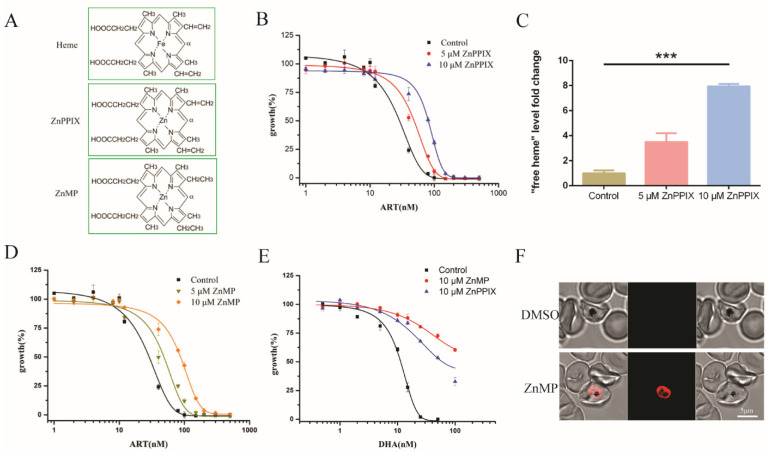
Heme analogues disrupted heme homeostasis and increased EC_50_ of artemisinin in *P. falciparum*. (**A**) Chemical structures of heme, ZnPPIX and ZnMP. (**B**) EC_50_ values of artemisinin in *P. falciparum* were increased by more than twofold in the presence of 10 μM ZnPPIX. (**C**) ZnPPIX elevated the free heme content by more than sixfold versus the control. (**D**) EC_50_ values of artemisinin were increased in the presence of ZnMP. (**E**) ZnPPIX and ZnMP decreased sensitivity of parasites to DHA. Data are mean ± SD of three experiments. (**F**) An amount of 20 μM ZnMP could efficiently reach into the parasite within an hour. The data were analyzed by GraphPad Prism. *** *p*-value < 0.0005.

**Figure 5 molecules-27-01755-f005:**
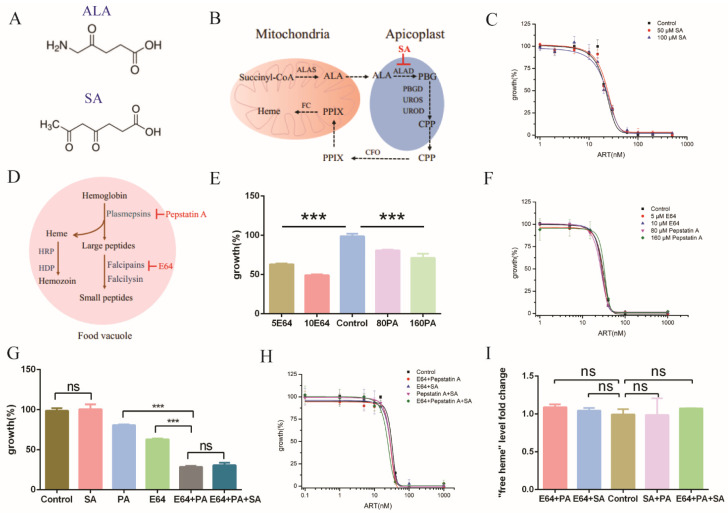
Inhibiting heme biosynthesis and/or hemoglobin degradation had little effect on the effect of artemisinin. (**A**) Chemical structure of ALA and SA. (**B**) Heme de novo biosynthesis pathway in *Plasmodium falciparum.* (**C**) EC_50_ of artemisinin in *P. falciparum* was not changed by the addition of SA. (**D**) Hemoglobin degradation and hemozoin formation pathway in *P. falciparum.* (**E**) Amounts of 5 μM/10 μM E64 and 80 μM/160 μM pepstatin A inhibited the growth of parasites after 72 h incubation. (**F**) The EC_50_ value of artemisinin remained unaltered in the presence of cysteine protease inhibitor E64 or aspartic protease inhibitor pepstatin A. (**G**) An amount of 5 μM E64 was synergistic or additive with 80 μM pepstatin A on malaria growth inhibition. (**H**) Combinatory use of inhibitors (5 μM E64, 80 μM pepstatin A and 50 μM SA) still produced little effect on artemisinin after 72 h incubation. When three inhibitors were used together, a very mild or marginal effect on the sensitivity of artemisinin was seen. (**I**) Free heme levels in *P. falciparum* were not significantly changed with inhibitors (5 μM E64, 80 μM Pepstatin A and 50 μM SA) after 20 h incubation. Data are mean ± SD of three experiments. The data were analyzed by GraphPad Prism. *** *p*-value < 0.0005, and ns = no significant change (*p*-value > 0.05).

**Figure 6 molecules-27-01755-f006:**
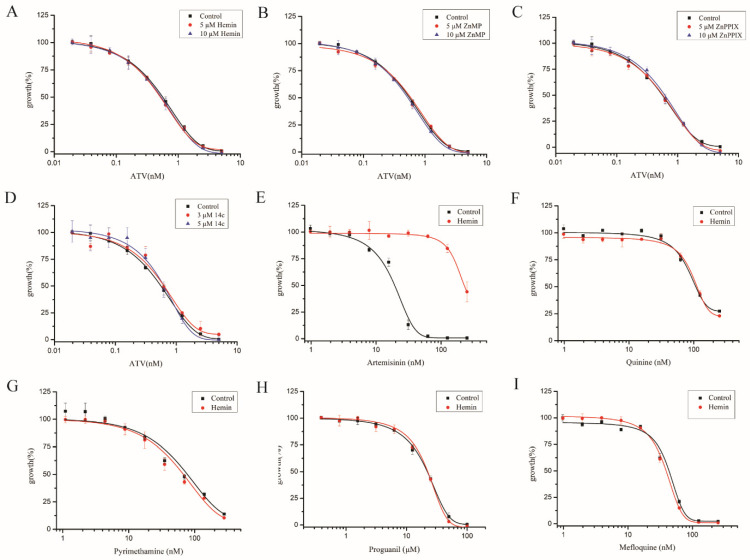
Heme had no effect on other antimalarial drugs. Parasites’ sensitivity to atovaquone had no significant change with 10 μM hemin (**A**), ZnPPIX (**B**), ZnMP (**C**) and 14c (**D**). Hemin decreased the sensitivity of artemisinin (**E**) but not antimalarial drugs quinine (**F**), pyrimethamine (**G**), mefloquine (**H**) and proguanil (**I**). ATV: atovaquone. Data are mean ± SD of three experiments.

## Data Availability

The data presented in this study are available on request from the corresponding author.

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
