# Peer review of "The Antagonizing Role of Heme in the Antimalarial Function of Artemisinin: Elevating Intracellular Free Heme Negatively Impacts Artemisinin Activity in Plasmodium falciparum"

_molecules, 2022, doi:10.3390/molecules27061755_

Round 1

Reviewer 1 Report

The manuscript by Zhu P. and Zhou B. presented a study evaluating the antimalarial effect of artemisinin under different levels of free heme in Plasmodium culture. The data are interesting and show a compromising effect of higher doses of heme in artemisinin action, and suggests a complex heme homeostasis pathway in Plasmodium. There are several studies related to the characterization of the complex between the heme group and artemisinin (uncited references: Ma, W. J. et al 2021 J. Biol Chem; Araujo JQ et al 2008 Bioorg Med Chem), however unsolved questions remain in the matter and the current manuscript presents some data that could contribute to clarifying the heme cell metabolism and raising novel questions.

Is necessary detailed information about the inhibition assays. It is not clear in methods section. For example: where are protease inhibitors (line 405 or 415)?? The concentrations used are missing in the text, are only mentioned in fig 5F, is similar for all assays??

Line 405Synchronized ring stage parasites were incubated with antimalarial drugs and chemicals, at 2% hematocrit and 1% starting parasitemia, for 72 h.”

Line 415Synchronized ring-stage parasites at 5% parasitemia and 3% hematocrit were cultured with DMSO or hemin analogues or inhibitors for 20 h.”

I also suggest a detailed incubation, concentrations info in figure legends.

The authors mentioned (line 245): “When heme levels in P. falciparum were measured, to our amazement, free heme levels in P. falciparum were not significantly altered even after the addition of multiple inhibitors (Figure5I).” It is really surprising that control was similar to the protease inhibitors group. To support the statement: “These results indicated that the free heme level inside parasites is robustly controlled,”. I suggest evaluating the hemozoin level or cell morphology (ex: DV size) because the effect of impaired source of heme and aminoacids with irreversible protease inhibitors heavily compromise cell metabolism.  I think it is very important for inclusion in figure 5 or supplemental material a gallery of giemsa images from each group of Figure 5I (what concentration and incubation period are used here?).

Could be the incubation period for protease inhibitor isn’t sufficient for observing changes in heme. For example, falcipain-2 presents higher activity after 24hs (Sijwali PS and Rosenthal PJ, 2004 PNAS; Prasad, R. et al 2013 Plos One), if incubation used was the initial 20hs can be not enough for E64 effect. In this case, an incubation window between 24-48hs is better for inhibitor treatment.

The English grammatical revision by a native speaker will benefit the text.

Minor concerns

As an example, for a required careful writing review of manuscript among others:

Page 2 line 44: “artemsinin”

Page 9 line 285: “quine

Page 10 line 355: “artemsinin

Author Response

Comments and Suggestions for Authors

The manuscript by Zhu P. and Zhou B. presented a study evaluating the antimalarial effect of artemisinin under different levels of free heme in Plasmodium culture. The data are interesting and show a compromising effect of higher doses of heme in artemisinin action, and suggests a complex heme homeostasis pathway in Plasmodium. There are several studies related to the characterization of the complex between the heme group and artemisinin (uncited references: Ma, W. J. et al 2021 J. Biol Chem; Araujo JQ et al 2008 Bioorg Med Chem), however unsolved questions remain in the matter and the current manuscript presents some data that could contribute to clarifying the heme cell metabolism and raising novel questions.

Is necessary detailed information about the inhibition assays. It is not clear in methods section. For example: where are protease inhibitors (line 405 or 415)?? The concentrations used are missing in the text, are only mentioned in fig 5F, is similar for all assays??

Response: We added some extra info in the Methods and legends. Most concentrations are labelled in the figures.

Line 405 “Synchronized ring stage parasites were incubated with antimalarial drugs and chemicals, at 2% hematocrit and 1% starting parasitemia, for 72 h.”

Response: Now it is changed to “Synchronized ring stage parasites were incubated with antimalarial drugs and chemicals(hemin, 14c, 14d, ZnPPIX, ZnMP, E64, Pepstatin A, SA), at 2% hematocrit and 1% starting parasitemia, for 72 h.”

Line 415 “Synchronized ring-stage parasites at 5% parasitemia and 3% hematocrit were cultured with DMSO or hemin analogues or inhibitors for 20 h.”

Response: It is changed to “Synchronized ring-stage parasites at 5% parasitemia and 3% hematocrit were cultured with DMSO or hemin analogue ZnPPIX or inhibitors(14c, 14d, E64, Pepstatin A, SA) for 20 h.”

I also suggest a detailed incubation, concentrations info in figure legends.

Response: Thank you for your suggestion and we have added detailed information in the revised figure legends, such as Figure 5E “5 μM/10 μM E64 and 80 μM/160 μM Pepstatin A could inhibit the growth of parasites after 72 h incubation.”

The authors mentioned (line 245): “When heme levels in P. falciparum were measured, to our amazement, free heme levels in P. falciparum were not significantly altered even after the addition of multiple inhibitors (Figure5I).” It is really surprising that control was similar to the protease inhibitors group. To support the statement: “These results indicated that the free heme level inside parasites is robustly controlled,”. I suggest evaluating the hemozoin level or cell morphology (ex: DV size) because the effect of impaired source of heme and aminoacids with irreversible protease inhibitors heavily compromise cell metabolism.  I think it is very important for inclusion in figure 5 or supplemental material a gallery of giemsa images from each group of Figure 5I (what concentration and incubation period are used here?).

Response: We showed hemoglobin level, free heme level and hemozoin level in parasites with protease inhibitors in Supplementary Figure 6. Protease inhibitor E64 increased hemoglobin level(Supplementary Figure 6B) by inhibiting falcipain hemoglobinases, but it did not affect the free heme level (Supplementary Figure 6C) or hemozoin level(Supplementary Figure 6D). This suggests that intraparasidic heme levels could be regulated by multiple pathways. When we used multiple inhibitors to inhibit heme-related pathway, we were unable to decrease free heme levels in parasites. In addition, these inhibitors could not alter hemozoin level in the parasite either (results shown below).

Could be the incubation period for protease inhibitor isn’t sufficient for observing changes in heme. For example, falcipain-2 presents higher activity after 24hs (Sijwali PS and Rosenthal PJ, 2004 PNAS; Prasad, R. et al 2013 Plos One), if incubation used was the initial 20hs can be not enough for E64 effect. In this case, an incubation window between 24-48hs is better for inhibitor treatment.

Response: We measured sensitivity of artemisinin after 72 h incubation, not 20 h, sorry for this confusion.

The English grammatical revision by a native speaker will benefit the text.

Minor concerns

As an example, for a required careful writing review of manuscript among others:

Response: Sorry for the typos and we have carefully checked the text and corrected in the revision. manuscript.

Page 2 line 44: “artemsinin”

Response: Corrected.

Page 9 line 285: “quine”

Response: Corrected.

Page 10 line 355: “artemsinin”

Response: Corrected.

Reviewer 2 Report

The authors investigate the role of heme for the antiplasmodial activity of artemisinin and dihydroartemisinin against P. falciparum in vitro.

Intracellular heme was increased by different experiments and the IC50 of artemisinin evaluated.

In addition, different inhibitors were also tested, experiments were also controlled with other antimalarials

Overall the authors describe a very interesting topic, and they performed a wealth of experiments

However, the work could be more clearly structured to make it more easy for the reader to follow.

Many different experiments are performed, and it is not always clear why the different experiments are performed, and what is the hypothesis behind the experiments.

It would also already help, if the abstract was more structured, and gives a more detailed description of the different experiments performed and the results obtained

Some minor comments:

Title: please change efficacy to a different work (e.g., activity)

Line 31 -  malaria is caused by parasites of the genus Plasmodium

Line 33 - and so on – please revise

Line 36 -  artemisinin revise

Line 43 - artemisinin typo

Line 44 – do not always lead… – not well explained - please refer to the ring stage survival assay and that this kind of resistance cannot be seen in the standard drug assay.

Line 47 and the following – please explain the differences between iron, ferrous iron, heme, hemin

Figure 1 C – your control pictures for 50 nM ART and 200 nM ART do not show any parasite… how does it look like after a shorter incubation

Line 79 – replace efficacy by activity or sth similar

In line 83 – effectiveness – I propose to always use activity. Efficacy and effectiveness are defined terms in clinical trials.

Line 98 – supressing parasitemia – please define more clearly what you mean by this. Does it inhibit that the parasites multiplies, so schizonts rupture and merozoites infect new cells?

Line 101 – sensitivity…of the parasite to DHA

Line 165 explain why EGFP is a control

393 (compounds sometimes with small letters, sometimes capital letters. Give more details to compounds

Author Response

Comments and Suggestions for Authors

The authors investigate the role of heme for the antiplasmodial activity of artemisinin and dihydroartemisinin against P. falciparum in vitro.

Intracellular heme was increased by different experiments and the IC50 of artemisinin evaluated.

In addition, different inhibitors were also tested, experiments were also controlled with other antimalarials

Overall the authors describe a very interesting topic, and they performed a wealth of experiments

However, the work could be more clearly structured to make it more easy for the reader to follow.

Many different experiments are performed, and it is not always clear why the different experiments are performed, and what is the hypothesis behind the experiments.

It would also already help, if the abstract was more structured, and gives a more detailed description of the different experiments performed and the results obtained

Some minor comments:

Title: please change efficacy to a different work (e.g., activity)

Response: Thank you for your suggestions and we changed title to “The antagonizing role of heme in the antimalarial function of artemisinin: elevating intracellular free heme negatively impacts artemisinin activity in Plasmodium falciparum”.

Line 31 -  malaria is caused by parasites of the genus Plasmodium

Response: Done as suggested.

Line 33 - and so on – please revise

Response: Thank you, we deleted “and so on” in the revised manuscript.

Line 36 -  artemisinin revise

Response: Corrected.

Line 43 - artemisinin typo

Response: Corrected.

Line 44 – do not always lead… – not well explained - please refer to the ring stage survival assay and that this kind of resistance cannot be seen in the standard drug assay.

Response: Thank you for your suggestions. We cited other papers reporting the ring stage survival of the mutant.

Line 47 and the following – please explain the differences between iron, ferrous iron, heme, hemin

Response: These are noted now in the revised manuscript. Iron is Fe, ferrous iron is Fe2+, and hemin is the oxidized heme.

Figure 1 C – your control pictures for 50 nM ART and 200 nM ART do not show any parasite… how does it look like after a shorter incubation

Response: This is a pic after 72 h incubation. Parasites are suppressed with 50 and 200 nM ART.

Line 79 – replace efficacy by activity or sth similar

Response: Done as suggested.

In line 83 – effectiveness – I propose to always use activity. Efficacy and effectiveness are defined terms in clinical trials.

Response: Done as suggested.

Line 98 – supressing parasitemia – please define more clearly what you mean by this. Does it inhibit that the parasites multiplies, so schizonts rupture and merozoites infect new cells?

Response: As suggested, we changed “Hemin antagonized with artemisinin in suppressing the parasitemia of P. falciparum” to “Hemin antagonized with artemisinin in inhibiting the parasite multiplication, suppressing the parasitemia of P. falciparum” in the revised manuscript.

Line 101 – sensitivity…of the parasite to DHA

Response: Corrected as suggested.

Line 165 explain why EGFP is a control

Response: The experiment is to show that expression of HO can reduce the heme level and thus concomitantly increases artemisinin potency. EGFP is an unrelevant protein to HO. If HO expression has the effect while EGFP has not, this indicates that the observation is specific to HO’s activity-in this case, its activity to degrade heme. We added a corresponding note in the revised manuscript.

393 (compounds sometimes with small letters, sometimes capital letters. Give more details to compounds

Response: Thank you for your suggestions and we have corrected to make them consistent in the revised manuscript. Brief introduction to the compounds, when applicable, was added.